# GPR4 Knockout Attenuates Intestinal Inflammation and Forestalls the Development of Colitis-Associated Colorectal Cancer in Murine Models

**DOI:** 10.3390/cancers15204974

**Published:** 2023-10-13

**Authors:** Mona A. Marie, Edward J. Sanderlin, Alexander P. Hoffman, Kylie D. Cashwell, Swati Satturwar, Heng Hong, Ying Sun, Li V. Yang

**Affiliations:** 1Department of Internal Medicine, Brody School of Medicine, East Carolina University, Greenville, NC 27834, USA; mona.marie@duke.edu (M.A.M.);; 2Department of Pathology, Brody School of Medicine, East Carolina University, Greenville, NC 27834, USA; 3Department of Pathology, Wake Forest University, Winston-Salem, NC 27157, USA

**Keywords:** inflammation, cancer, inflammatory bowel disease (IBD), colitis-associated colorectal cancer, GPR4, G protein-coupled receptor (GPCR), angiogenesis, fibrosis, knockout mice

## Abstract

**Simple Summary:**

Inflammatory bowel disease (IBD), including ulcerative colitis and Crohn’s disease, is a debilitating condition with chronic inflammation in the digestive tract. Patients with IBD are at higher risk of developing colitis-associated colorectal cancer (CAC) compared with the general population. The etiology of IBD is not well understood, but both genetic and environmental factors have been implicated. In this study, we investigated the role of the pH-sensing GPR4 receptor in colitis and CAC mouse models. GPR4 knockout alleviated intestinal inflammation, reduced tumor angiogenesis, and impeded CAC development. Our data suggest that the inhibition of GPR4 may be explored as a potential therapeutic approach for IBD treatment and CAC prevention.

**Abstract:**

GPR4 is a proton-sensing G protein-coupled receptor highly expressed in vascular endothelial cells and has been shown to potentiate intestinal inflammation in murine colitis models. Herein, we evaluated the proinflammatory role of GPR4 in the development of colitis-associated colorectal cancer (CAC) using the dextran sulfate sodium (DSS) and azoxymethane (AOM) mouse models in wild-type and GPR4 knockout mice. We found that GPR4 contributed to chronic intestinal inflammation and heightened DSS/AOM-induced intestinal tumor burden. Tumor blood vessel density was markedly reduced in mice deficient in GPR4, which correlated with increased tumor necrosis and reduced tumor cell proliferation. These data demonstrate that GPR4 ablation alleviates intestinal inflammation and reduces tumor angiogenesis, development, and progression in the AOM/DSS mouse model.

## 1. Introduction

Colitis-associated colorectal cancer (CAC) is driven by chronic and recurring mucosal inflammation observed in inflammatory bowel disease (IBD) patients [1]. Patients with IBD have an increased risk of developing CAC when compared with the general population [2]. In this context, colorectal cancer (CRC) can develop and progress through the chronic inflammation–dysplasia–carcinoma axis [1]. Interestingly, severe intestinal inflammation is associated with decreased colon luminal pH in IBD patients [3,4,5]. Tissue acidosis is characterized by reduced extracellular pH and is a hallmark of chronic inflammation and cancer. As such, solid tumors are characterized by an acidic tumor microenvironment [6,7,8]. Local tissue acidification of inflamed colonic tissues has also been observed in a DSS-induced colitis mouse model [9]. Inflamed colon segments excised from DSS-treated mice were more acidic than non-inflamed colon segments, as measured by the pH indicator SNARF-4F 5-(and-6)-carboxylic acid [9].

A family of proton-sensing G protein-coupled receptors (GPCRs) capable of sensing changes in extracellular pH have recently been implicated in regulating intestinal inflammation. The pH-sensing GPCR family includes GPR4, GPR65, and GPR68, which are activated via the protonation of histidine residues on the receptor extracellular domains [7,10,11,12,13,14,15]. GPR4 is predominately expressed in vascular endothelial cells, among other cell types [16,17,18,19,20,21,22]. Our group has demonstrated that GPR4 activation via the acidic microenvironment augments endothelium adhesiveness and increases the expression of endothelial proinflammatory molecules, such as IL-8, CXCL2, COX-2, VCAM-1, and E-selectin [17,23]. Moreover, intestinal fibrosis is a serious complication in IBD [24,25]. GPR4 expression is positively correlated with fibrogenic gene expression in highly fibrotic intestine lesions obtained from Crohn’s disease (CD) patients [20]. In addition to proinflammatory, profibrotic, and endothelial cell activation roles, GPR4 has been implicated in both physiological and pathological angiogenesis [19,20,26,27,28]. Studies have demonstrated a role for GPR4 in promoting angiogenesis and tumor growth in the porous tissue implant and orthotopic tumor models [28]. GPR4 expression is associated with increased angiogenesis in hepatocellular, head and neck, breast, and colorectal cancers [27,28,29]. Increased GPR4 expression has been observed in colorectal tumors compared with adjacent normal tissue and is associated with decreased overall survival in patients [30]. Therefore, GPR4 may contribute to tumorigenesis in the colon by reinforcing the inflammatory and angiogenic processes in IBD. Herein, using wild-type and GPR4 knockout mouse CAC models, we investigated the role of GPR4 in inflammation-driven colorectal cancer.

## 2. Materials and Methods

### 2.1. Ethics Statement

These mouse experiments were approved by the Institutional Animal Care and Use Committee of East Carolina University in accordance with the Guide for the Care and Use of Laboratory Animals (The National Academies Press). 

### 2.2. Dextran Sulfate Sodium (DSS)-Induced Colitis Mouse Model

Nine-week-old male and female wild-type (WT) and GPR4 knockout (KO) mice were used in the experiments. GPR4-deficient mice were generated as previously described and were backcrossed into the C57BL/6 background for 11 generations [19]. Animals were maintained under specific pathogen-free conditions and were free from Helicobacter, Citrobacter rodentium, and norovirus. Colitis was induced using 3% (*w*/*v*) colitis-grade dextran sulfate sodium (DSS) with molecular weight 36,000–50,000 Da (Lot# Q1408, MP Biomedical, Solon, OH, USA) within the drinking water of the mice. The 3% DSS solution or water was provided to mice ad libitum, as previously described [16,31,32]. Briefly, mice were given 3% DSS for 4 cycles. Each cycle constituted 5 days of 3% DSS followed by 2 days of water. Following the fourth cycle, water was switched back to 3% DSS for 2 final days. Mouse body weight and clinical phenotype scores were measured each day [33]. 

### 2.3. Azoxymethane (AOM) and Dextran Sulfate Sodium (DSS)-Induced Colitis-Associated Colorectal Cancer Mouse Model (CAC)

Nine-to-twelve-week-old male and female WT and GPR4 KO mice were used in these experiments. Mice were backcrossed 13 generations into the C57BL/6 background. CAC mouse model was induced as previously described [33,34,35]. Briefly, a single i.p. injection (10 mg/kg) of azoxymethane (AOM, product# A5486, Sigma-Aldrich, Saint Louis, MO, USA) followed by 4% (*w*/*v*) DSS (Lot# Q5229 and S0948, MP Biomedical, Solon, OH, USA) in drinking water was used to induce colitis-associated colorectal cancer in mice. The mice were given 4% DSS in water for 3 cycles. Each cycle constituted 5 days of 4% DSS followed by 16 days of water. Following the third cycle, mice were given water for the remaining period (37 days) to the endpoint on the 14th week (100 days). 

### 2.4. Mouse Clinical Phenotype Scoring

Assessment of colitis severity in WT and GPR4 KO mice treated with 3% DSS or AOM/ 4% DSS was performed as previously described [33]. Colitis severity was determined using the clinical parameters of body weight loss and fecal score [16]. Disease activity index, represented by body weight loss percentage, fecal score, colon shortening, and mesenteric lymph node expansion, was measured to assess inflammation. Feces were collected from mice and assessed for presence of blood and consistency. Fecal scoring system consisted of the following: 0 = normal, dry, firm pellet; 1 = formed soft pellet with negative hemoccult test; 2 = formed soft pellet with positive hemoccult test; 3 = formed soft pellet with visual blood; 4 = liquid diarrhea with visual blood; 5 = bloody mucus and no colonic fecal content upon necropsy. Presence of micro-blood content was measured using the Hemoccult Single Slides screening test (Beckman Coulter, Brea, CA, USA). 

### 2.5. Mouse Tissue Collection, Evaluation, and Processing

At endpoint, mice were euthanized followed by necropsy; colon and mesenteric lymph nodes were collected, evaluated, and processed, as previously described [16,33]. Colon length was measured from the ileocecal junction to anus. Then, colon was separated from cecum, and phosphate buffer saline (PBS) was used to clear it of fecal content; then, it was opened along the anti-mesenteric border. Colon tissue was fixed with 10% buffered formalin and cut evenly into distal, middle, and proximal sections for histologic evaluation. The mesenteric lymph node and tumor volumes were assessed with a caliper using the formula (length × width^2^) π/6 using a dissecting scope for visualization and then fixed with 10% buffered formalin and collected for histological analysis. 

### 2.6. Histopathological Analysis 

Subsequently, 5 µm sections of distal, middle, and proximal colon tissue segments were obtained from WT and GPR4 KO mice and stained with hematoxylin and eosin (H&E) for histological analysis. Sample identification was concealed during histopathological analysis for unbiased evaluation. Board-certified medical pathologists evaluated colitis histopathological features, including inflammation, crypt damage, edema, architectural distortion, and leukocyte infiltration in a blinded manner, as previously described [16,31,32]. Each parameter was scored and multiplied by a factor corresponding to total tissue percentage of disease involvement. Additional scoring criteria for colonic fibrosis were evaluated as previously described [36]. Briefly, colon segments of WT and GPR4 KO mice treated with DSS were stained with picrosirius red for fibrosis analysis and graded in a blinded manner for pathological fibrosis. Severity of fibrosis was included as one of the parameters of the histopathological score. WT and GPR4 KO AOM/DSS-treated mouse intestinal tumors in the colons were assessed to differentiate dysplasia and adenocarcinoma lesions using the histologic criteria: neoplastic cells in adenocarcinoma invade through the muscularis mucosa into the submucosa, whereas neoplastic cells in dysplasia are confined within the basement membrane of the mucosa.

### 2.7. Tumor Necrotic Area Quantification 

Hematoxylin and eosin (H&E) stain was performed on formalin-fixed, paraffin-embedded tumor sections to assess necrosis in WT and GPR4 KO AOM/DSS mouse colons. Percent of necrotic area per field of view (FOV) was measured using the ImageJ software (v.1.53k) in a blinded manner. Briefly, images were taken of each tumor to capture the total tumor area using a microscope (Axio Imager M2, Carl Zeiss Inc., White Plains, NY, USA). Percent of necrosis per FOV was calculated using the following equation: % necrotic area = (sum of necrotic areas/total area of FOV) × 100. 

### 2.8. Immunohistochemistry

Colon tissues were embedded in paraffin, and serial 5 µm sections were evaluated for immunohistochemical analysis, as previously described [16]. Briefly, antigen retrieval was performed on colon sections followed by endogenous peroxidase blocking. Endogenous biotin, biotin receptors, and avidin binding sites were blocked using an avidin/biotin blocking kit (Vector Laboratories, California, CA, USA) followed by normal serum blocking and stained with anti-green fluorescent protein (GFP) (Abcam, ab6673, Cambridge, MA, USA), anti-F4/80 (Cell Signaling Technology, #70076, Danvers, MA, USA), anti-CD4 (Abcam #ab183685, Waltham, MA, USA), anti-CD8 (Abcam, #ab203035, Waltham, MA, USA),anti-cleaved caspase-3 (Cell Signaling Technology, #9664S, Danvers, MA, USA), anti-CD31 (Cell Signaling Technology, #77699S, Danvers, MA, USA), and anti-Ki67 (Abcam, #ab15580, Waltham, MA, USA) primary antibodies. The IHC detection system VECTASTAIN^®^ Elite ABC-HRP Kit, Peroxidase (Rabbit IgG) (Vector Laboratories, California, CA, USA), and VECTASTAIN^®^ Elite ABC-HRP Kit, Peroxidase (Goat IgG) (Vector Laboratories, California, CA, USA) were used. Following the addition of secondary antibody, DAB (3,3′-diaminobenzidine) incubation was performed for HRP detection. Pictures were taken using the Zeiss Axio Imager M2 microscope. 

### 2.9. Microvessel Density Quantification

CD31^+^ immunohistochemistry stain was used as a marker for endothelial cells forming blood vessels in the WT and GPR4 KO AOM/DSS colon tumor sections. Image J (v.1.53k) was used for manual quantification of individual blood vessels/field of view (FOV). In total, 5–8 images were captured per tumor, and blood vessel numbers were averaged. Images were analyzed in a blinded manner. Data are represented as the averaged tumor blood vessel number/FOV for the two groups, WT and GPR4 KO AOM/DSS.

### 2.10. Tumor Proliferation Quantification

Ki67 immunohistochemistry stain was used as a proliferation marker in the WT and GPR4 KO AOM/DSS colon tumor sections. The Fiji (v.1.53q) software was used for analysis, as previously described [37]. Briefly, 4–8 images were captured per tumor and analyzed in a blinded manner. Percent positive cells were calculated by dividing the number of Ki67-positive (brown) cells by the total number of cells (blue nuclei)/FOV, multiplied by 100. Then, all FOV percentages were averaged to produce a percentage per tumor. 

### 2.11. Quantitative Reverse Transcription Polymerase Chain Reaction (qRT-PCR)

Real-time PCR was performed as previously described [16]. Crohn’s and ulcerative colitis cDNA arrays were purchased from Origene Technologies (catalog #CCRT102, Rockville, MD, USA) and utilized in real-time PCR analysis using specific primer–probes for human *GPR4*, *TNF-α*, *IFN-γ*, and *β-actin* [23,33]. cDNA array patient information was described in a previous report, as supplied by the vendor [16]. Furthermore, total RNA was isolated from the chronic DSS and control mouse colon tissues using the IBI extraction kit (catalog #IB47702, Fenton, MO, USA). RNA was reverse-transcribed into cDNA using the SuperScript IV reverse transcriptase (ThermoFisher Scientific, CA, USA). TaqMan qRT-PCR was performed to measure gene expression using specific primer–probes for mouse *Gpr4* (Mm00558777_s1), *Tnf-α* (Mm00443258_m1), *Ifn-γ* (Mm01168134_m1), *Cxcl2* (Mm00436450_m1), *Cox2/Ptgs2* (Mm00478374_m1), *Madcam1* (Mm00522088_m1), *E-selectin* (Mm00441278_m1), *Icam1* (Mm00516023_m1), and *Vcam1* (Mm01320970_m1). Gene expression was normalized to the housekeeping gene *18S rRNA* (Hs99999901_s1), and the relative gene expression was calculated using the 2^−∆∆Ct^ method.

### 2.12. Statistical Analysis 

All statistical analyses were performed using the GraphPad Prism (v.9.4.1) software. The unpaired *t*-test, the Mann–Whitney U test, and one-way ANOVA were used to compare differences between two groups (WT vs. GPR4 KO) or more than two groups. Linear regression and Pearson and Spearman correlation were used to correlate gene expression. Python 3 code was written using NumPy, Panda, and matplotlib to calculate the correlation and regression of mouse tumor blood vessel density, tumor volume, and necrosis area and then to graph the scatter plot. *p* < 0.05 is considered statistically significant.

## 3. Results

### 3.1. GPR4 Potentiates Intestinal Inflammation in the Chronic DSS-Induced Experimental Colitis Mouse Model

We used WT and GPR4 KO mice to characterize the role of GPR4 in the chronic DSS-induced colitis mouse model [38,39]. The model consists of four cycles of intestinal inflammatory insults. Mouse body weight and fecal blood and diarrhea were analyzed as disease activity indicators (Figure 1). The WT-DSS mice began to lose between 12–15% body weight following cycle one, whereas the GPR4 KO-DSS mice lost between 5–10% body weight throughout the cycles (Figure 1A). The fecal scores also indicated that the GPR4 KO-DSS mice were less clinically severe when compared with the WT-DSS mice, as the GPR4 KO-DSS mice had reduced fecal blood and diarrhea (Figure 1B). Upon completion of all four cycles of the chronic DSS-induced colitis, macroscopic disease indicators were collected such as mesenteric lymph node (MLN) enlargement and colon length measurements. MLN volume change due to inflammation was not different at this point of the study between the WT and GPR4 KO-DSS mice (Figure 1C). The colon length, however, indicated that the GPR4 KO-DSS mice had less colon shortening when compared with the WT-DSS mice (Figure 1D). Collectively, these results indicate that GPR4 potentiates disease severity in the chronic DSS-induced mouse model.

We then evaluated the degree of histopathology in the distal, middle, and proximal colon segments. Distinct parameters of colitis-associated histopathology were assessed, such as leukocyte infiltration, edema, crypt loss, and architectural distortion, to obtain a score of severity. The WT and GPR4 KO untreated water control mice displayed no observable histopathology (Figure 2A and Appendix A). The GPR4 KO-DSS mice had reduced histopathology when compared with the WT-DSS mice in their colon segments (Figure 2A,B).

Moreover, leukocyte infiltration was reduced in the colons of GPR4 KO-DSS mice when compared with WT-DSS mice (Figure 2C), which corroborates previous reports indicating that GPR4 can increase leukocyte infiltration into inflamed intestinal tissues by upregulating endothelial cell adhesion molecules [16]. Further immunohistochemical analyses showed that F4/80+ macrophages, CD4+ T cells, and CD8+ T cells were reduced in GPR4 KO-DSS mouse distal colons compared with WT-DSS (Appendix A).

Another distinct histopathological consequence is fibrosis in chronically inflamed intestinal tissues. We observed heightened fibrotic development in mice with chronic DSS-induced colitis. The distal colon segment displayed the highest degree of fibrosis with a progressive reduction in severity from the middle to the proximal colon. A significant reduction in fibrosis was observed in the GPR4 KO-DSS mice in the distal, middle, and proximal colon segments when compared with the WT-DSS mice (Figure 2D,E and Appendix A).

### 3.2. GPR4 Gene Expression Is Upregulated in Inflamed Colon Tissues and Positively Correlated with TNF-α and INF-γ Gene Expression

Previous studies have shown that GPR4 gene expression is upregulated in both ulcerative colitis (UC) and Crohn’s disease (CD) patient intestinal samples compared with normal intestinal tissue [16,18]. TNF-α and INF-γ are cytokines implicated in the inflammatory pathways of IBD [40]. Herein, we observed a positive correlation between the gene expression of *GPR4* with both *TNF-α* (*p* < 0.0001 and R = 0.5966) and *INF-γ* (*p* < 0.0001 and R = 0.5634) in IBD patient intestinal samples (Figure 3A,B). Furthermore, *GPR4* gene expression was upregulated three-fold in the WT-DSS mouse colon samples compared with the WT control (Figure 3C), consistent with upregulation of *GPR4* expression observed in other mouse colitis models and IBD patient samples [16,18,20,32]. *GPR4* gene expression was also correlated with *TNF-α* and *INF-γ* expression in the mouse colon samples (Figure 3D,E). When compared with the WT-DSS mouse colons, *Cox2* (*Ptgs2*) gene expression was significantly reduced in the GPR4 KO-DSS colons, and there was also a trend of reduced *Cxcl2*, *Madcam1*, and *E-selectin* gene expression in the GPR4 KO-DSS colons (Appendix A).

### 3.3. Genetic Deletion of GPR4 Reduces Disease Severity in CAC Induced by AOM/DSS in Mice

To further study the role of GPR4 in the development of colitis-associated colorectal cancer (CAC), we utilized the well-established AOM/DSS murine model [41,42]. Disease severity indicators were measured during the experiment by monitoring body weight loss and fecal blood and diarrhea scores. The WT AOM/DSS mice showed more significant body weight loss when compared with the GPR4 KO AOM/DSS mice starting at the end of the first cycle and throughout the second and third cycles. Both the WT and GPR4 KO AOM/DSS mice reached an average body weight loss of ~17% by day 9 during the first cycle, followed by a partial recovery in body weight loss. This body weight recovery was more significant in the GPR4 KO mice than the WT mice (Figure 4A). The severity of fecal blood and diarrhea scores was also significantly higher in the WT AOM/DSS mice in comparison with the GPR4 KO AOM/DSS mice (Figure 4B). At the endpoint of the experiment (day 100), macroscopic disease indicators were evaluated, such as final body weight, mesenteric lymph node expansion, and colon length shortening. The GPR4 KO AOM/DSS mice showed a significant increase in body weight recovery compared with the WT mice (Figure 4C). The mesenteric lymph node volume of the WT mice was larger than that of the GPR4 KO mice, indicating a more severe intestinal inflammation in the WT mice (Figure 4D). No significant differences were observed in colon length between the WT and the GPR4 KO AOM/DSS-treated mice (Figure 4E).

### 3.4. GPR4 Knockout Reduces Tumor Burden in the CAC Mouse Model

A significant increase in colon tumor number was observed in the WT AOM/DSS mice when compared with the GPR4 KO AOM/DSS mice (Figure 5A,B), suggesting GPR4 promotes CAC development. The WT mice had an average tumor number of 7.4, whereas the GPR4 KO mice had an average of 5.1, demonstrating a 45.1% increase in tumor number in the WT over GPR4 KO AOM/DSS mice (Figure 5B). Moreover, the volume of the detected tumors was 115 mm^3^ in the WT mice and 55 mm^3^ in the GPR4 KO AOM/DSS mice (Figure 5C). Histological analyses of the colon tissue sections revealed that colon dysplasia (Dys) and adenocarcinoma in situ (AIS) were induced in both the WT and GPR4 KO AOM/DSS mice (Figure 5D,E). Interestingly, the distribution of dysplasia and adenocarcinoma in the WT and GPR4 KO AOM/DSS mice showed a 9% decrease in adenocarcinomas in the GPR4 KO mice compared with the WT mice (85% vs. 94%, respectively) (Figure 5E). These data suggest the absence of GPR4 delays progression from low-grade dysplasia to adenocarcinoma in inflamed colon tissues.

### 3.5. GPR4 Knockout Increases Necrosis and Cell Death and Decreases Cell Proliferation in the Tumors of AOM/DSS Mice

Tumor necrosis is considered a common feature of solid tumors, which occurs as a consequence of nutrient and oxygen deprivation [43]. Areas in the tumors with apparent gaps resulting from transformed crypt loss were identified as necrotic areas, where remnants of nuclear and cytoplasmic debris were visualized with H&E stain (Figure 6A). The percentage of tumor necrosis areas between the WT and GPR4 KO AOM/DSS tumors was ~ 2.4-fold higher in the GPR4 KO AOM/DSS tumors compared with the GPR4 WT AOM/DSS tumors (Figure 6A,B). Using immunohistochemistry, we assessed the protein expression of cleaved caspase-3, a marker of programmed cell death, in WT and GPR4 KO AOM/DSS tumors. The expression of cleaved caspase-3 was predominantly detected in the necrotic tumor area in addition to positive epithelial cells scattered within the tumor (Figure 6C). Next, we evaluated Ki67 as a proliferation marker to assess tumor cell proliferation in the WT and GPR4 KO AOM/DSS mice. WT AOM/DSS tumors showed a more proliferative phenotype when compared with GPR4 KO AOM/DSS tumors (Figure 6D). The quantification of Ki67-positive cells revealed a ~two-fold decrease in proliferating tumor cells in GPR4 KO AOM/DSS mouse tumors (Figure 6E).

### 3.6. GPR4 Is Highly Expressed in the Tumor Blood Vessels of AOM/DSS Mice

We next characterized the expression profile of GPR4 in the AOM/DSS mouse tumor model. Immunohistochemistry was performed for green fluorescent protein (GFP), which functions as a surrogate marker for GPR4 expression in GPR4 KO mice. GPR4 KO mice were generated by replacing the GPR4 coding region with an internal ribosome entry site (IRES)-GFP cassette under the control of the endogenous GPR4 gene promoter, as previously described [16,19].

The GFP signal was detected in GPR4 KO AOM/DSS but not WT AOM/DSS colon tissues. In line with our previous observations [16], high levels of GFP expression were detected in the endothelial cells of tumor blood vessels (Figure 7A). Furthermore, using a double fluorescent stain with GFP and the endothelial marker CD31, we observed that GFP expression was predominately detected in the endothelial cells (ECs) of blood vessels in the tumor tissue of GPR4 KO AOM/DSS colons (Figure 7B).

### 3.7. GPR4 Deletion Decreases Angiogenic Blood Vessel Formation in the Tumors of AOM/DSS Mice

During tumorigenesis, angiogenesis is a fundamental process in tumor development and progression [44]. We analyzed the blood vessel density in the tumors of WT AOM/DSS and GPR4 KO AOM/DSS mice using immunohistochemistry. The endothelial marker CD31 was used to identify tumor blood vessels for counting. The deficiency of GPR4 in the GPR4 KO AOM/DSS mice caused a significant reduction in blood vessel density in the tumors of these mice by ~2.6 fold when compared with WT AOM/DSS mice (Figure 8A,B). Moreover, blood vessel density positively correlated with tumor volume and negatively correlated with tumor necrosis area in the WT AOM/DSS and GPR4 KO AOM/DSS mouse colon tumors (Figure 8C,D).

## 4. Discussion

In this study, we demonstrate that GPR4-driven intestinal inflammation can increase the development of colorectal tumorigenesis in a colitis-associated colorectal cancer mouse model. Our observations are in line with previous reports establishing a proinflammatory role for GPR4 in several systems, such as the brain, heart, kidney, lung, bone, skin, and gastrointestinal tract [16,18,19,20,32,45,46,47,48,49,50,51]. In addition, GPR4 has a protumorigenic role in hepatocellular, head and neck, breast, and colorectal cancers [27,28,29].

We and others previously elucidated the role of GPR4 in potentiating intestinal inflammation by using both GPR4 antagonists and global GPR4 knockout mice in colitis mouse models [16,18,20,32]. These studies demonstrated a proinflammatory role for GPR4 in colitis via the activation of the intestinal endothelium to facilitate immune cell recruitment and infiltration into inflamed intestinal tissues [16,18,20,32]. Furthermore, GPR4 mRNA expression levels are significantly increased in the inflamed intestinal tissues of both human IBD patients and in colitis mouse models when compared with normal tissues. These data implicate GPR4 in the pathophysiology of chronic intestinal inflammation [16,18].

Patients with IBD are at a ~2.4-fold higher risk of developing colitis-associated colorectal cancer (CAC) because of persistent and unresolved intestinal inflammation [2]. Local tissue acidosis exists in the inflammatory loci in part because of shifts in hypoxia, glycolytic cellular metabolism, the production of bacterial metabolic byproducts, and granulocyte respiratory bursts [6,7,8,52]. Increased proton concentrations in the extracellular milieu can activate GPR4 via the protonation of histidine residues on the extracellular domain and initiate the activation of downstream G protein pathways [53]. We and others previously demonstrated that GPR4 activation in endothelial cells elicits the production of inflammatory molecules such as TNF and NF-κB family members, chemokines, cytokines, and adhesion molecules and functionally mediates leukocyte adhesion to the activated endothelium [16,23,32,54]. In this study, we demonstrated that GPR4 increased intestinal inflammation in DSS chronic colitis and AOM/DSS colitis-associated colorectal cancer mouse models (Figure 1, Figure 2, Figure 3 and Figure 4). Furthermore, a positive correlation between elevated GPR4 expression with elevated TNF-α and IFN-γ expression in the intestinal tissues of IBD patients was observed (Figure 3). We previously showed that GPR4 antagonists in the DSS-induced colitis mouse model reduced TNF-α mRNA levels in intestinal tissues when compared with a vehicle control [32]. Several studies have linked the activation of TNF-α and NF-κB pathways to initiating the malignant transformation of intestinal epithelial cells [55,56]. These data are supported by studies employing the genetic and pharmacological blockade of TNF-α, which reduced the CAC tumor burden in mice [57,58]. Moreover, a trend of reduced *COX2* (*PTGS2*), *CXCL2*, *MADCAM1*, and *E-Selectin* gene expression was observed in GPR4 KO-DSS mouse colons when compared with WT-DSS colons (Appendix A). These data are consistent with previous observations of the GPR4-mediated regulation of inflammatory gene expression in several animal models [22,45,47,50,59]. Concordantly, COX2 (PTGS2) is overexpressed in the colonic mucosa of IBD patients and plays an important role in intestinal inflammation [60]. CXCL2 is a chemokine involved in leukocyte recruitment, while MADCAM1 and E-Selectin are vascular adhesion molecules important for leukocyte adhesion and extravasation. The reduced expression of these molecules in the GPR4 KO-DSS mouse colon is in line with a decrease in leukocyte infiltration observed in the GPR4 KO-DSS colon tissues (Figure 2 and Appendix A). In this context, GPR4 can contribute to chronic inflammation and tumor development, which can feedforward in the inflammation–dysplasia–carcinoma axis, resulting in CAC development [1].

Interestingly, it has also been reported that GPR4 expression is increased in hepatocellular carcinoma, head and neck cancer, and CRC tissues compared with normal tissues [27,29,30]. Additionally, hepatocellular and CRC patients with high GPR4 expression have shown poor prognosis and decreased survival [29,30]. Furthermore, a significant reduction in breast and CRC tumor growth in GPR4 knockout mice and in vitro CRC cell proliferation caused by GPR4 knockdown has been observed [28,30]. These observations are in line with our findings, in which we observed reduced colon tumor burden in GPR4 KO mice compared with WT mice in the AOM/DSS mouse model (Figure 5). While the AOM/DSS mouse model was used in this study, CRC can be induced in mice using AOM or DSS alone, in which tumors take a longer time to develop [61,62]. These mouse models can be used in future research to tease out the effects of AOM and DSS.

Another protumorigenic role for GPR4 in vascular endothelial cells is likely related to angiogenesis [27,28,29]. GPR4 has been shown to promote angiogenesis by regulating the VEGF pathway in both ischemic tissues and cancer tissues [20,28,49]. One study showed that upregulating GPR4 enhanced the angiogenesis of endothelial progenitor cells (EPCs) isolated from chronic artery disease patients through VEGFA/STAT3 activation. The study also showed increased blood flow in a mouse hindlimb ischemia model following the injection of GPR4-overexpressing EPCs [49]. Another study demonstrated the reduced growth of orthotopic breast cancer and colon cancer allografts in GPR4 KO mice due to decreased angiogenesis [28]. Consistently, GPR4 KO mouse tumors showed reductions in tumor angiogenesis when compared with WT tumors in the AOM/DSS mouse model (Figure 8). We previously characterized the expression of GPR4 in the intestinal tissues of both the naïve and DSS-induced colitis disease state using GPR4 KO mice with a GFP knock-in [16]. In this system, GFP is under the regulation of the endogenous GPR4 promoter and can be used as a surrogate marker for endogenous GPR4 expression. We observed that GPR4 is predominately expressed in the vascular endothelial cells of the arteries, veins, and microvessels of both the cecum and colon [16]. In this study, we employed the double immunolabeling of GFP and CD31 to detect GPR4 expression in tumor vascularization (Figure 7). We found that GPR4 is expressed in the vascular endothelial cells of colorectal tumors in AOM/DSS mice and may be involved in tumor vascularization. As shown in previous studies, GPR4 may regulate angiogenesis through the VEGF/VEGFR pathway, STAT3, and the production of proangiogenic factors [20,27,28,49]. Our observations in this study suggest that the decrease in microvessel density in tumors of GPR4 KO AOM/DSS mice is associated with increased tumor cell death and reduced tumor cell proliferation caused by the prevention of adequate tumor vascularization (Figure 6 and Figure 8). Collectively, the genetic deletion of GPR4 likely halts colorectal cancer development by dampening chronic intestinal inflammation and impeding tumor angiogenesis.

The therapeutic benefit of GPR4 inhibition has been described as reducing inflammation, pain, and angiogenesis [22,45,47,50,54,59,63,64]. We and others have shown the therapeutic value of GPR4 inhibitors in IBD pre-clinical mouse models [20,32]. Additional therapeutic benefits of GPR4 antagonism have been reported in tissue ischemia, myocardial infarction, chronic obstructive pulmonary disease (COPD), and osteoarthritis models, reducing proinflammatory molecules such as VCAM-1, E-selectin, IL-17, interferon-γ, TNF-α, IL-1β, IL-6, inducible nitric oxide synthase (iNOS), nitric oxide (NO), cyclooxygenase 2 (COX2), prostaglandin E2 (PGE2), Mucin5AC, matrix metalloprotease (MMP)-9, MMP-12, and NF-κB [22,45,47,50,59]. Based on these research findings, we propose that GPR4 inhibition (e.g., via small-molecule antagonists, siRNAs, antisense oligonucleotides, and antibodies) can be explored as a potential therapeutic approach for colitis treatment and CAC prevention in IBD patients by alleviating chronic intestinal inflammation and inhibiting pathological angiogenesis.

## 5. Conclusions

Our results demonstrate that the knockout of GPR4 alleviates intestinal inflammation and impedes the development of colitis-associated colorectal cancer in mouse models. Furthermore, GPR4 knockout reduces tumor angiogenesis and growth. The data suggest that the inhibition of GPR4 may be exploited as a potential approach to anti-inflammatory and anti-cancer therapy.

## Figures and Tables

**Figure 1 cancers-15-04974-f001:**
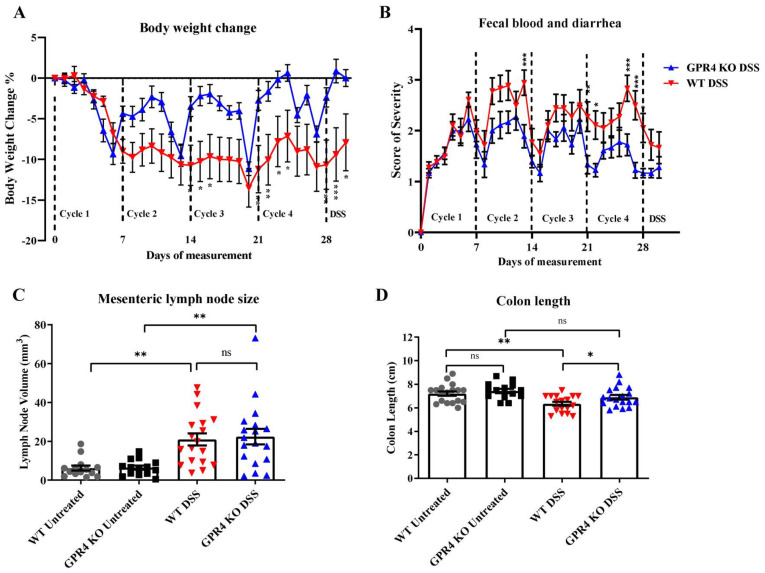
Disease activity indicators of chronic colitis induction in wild-type (WT) and GPR4 knockout (KO) mice. DSS-induced inflammation was assessed in WT-DSS and GPR4 KO-DSS mice. WT-DSS mice presented elevated inflammatory indicators compared with GPR4 KO-DSS mice. Clinical phenotypes of intestinal inflammation such as (**A**) body weight loss and (**B**) fecal blood and diarrhea were assessed. Macroscopic disease parameters such as (**C**) mesenteric lymph node expansion and (**D**) colon shortening were also recorded. Data are presented as the mean ± SEM, and statistical significance was determined using an unpaired *t*-test between the WT-DSS and GPR4 KO-DSS groups or the one-way ANOVA test between four groups: the WT control, WT-DSS, the GPR4 KO control, and GPR4 KO-DSS. WT control (n = 18), WT-DSS (n = 17), GPR4 KO control (n = 14), and GPR4 KO-DSS (n = 18) mice were used for all experiments (* *p* < 0.05, ** *p* < 0.01, *** *p* < 0.001, *ns*: not significant).

**Figure 2 cancers-15-04974-f002:**
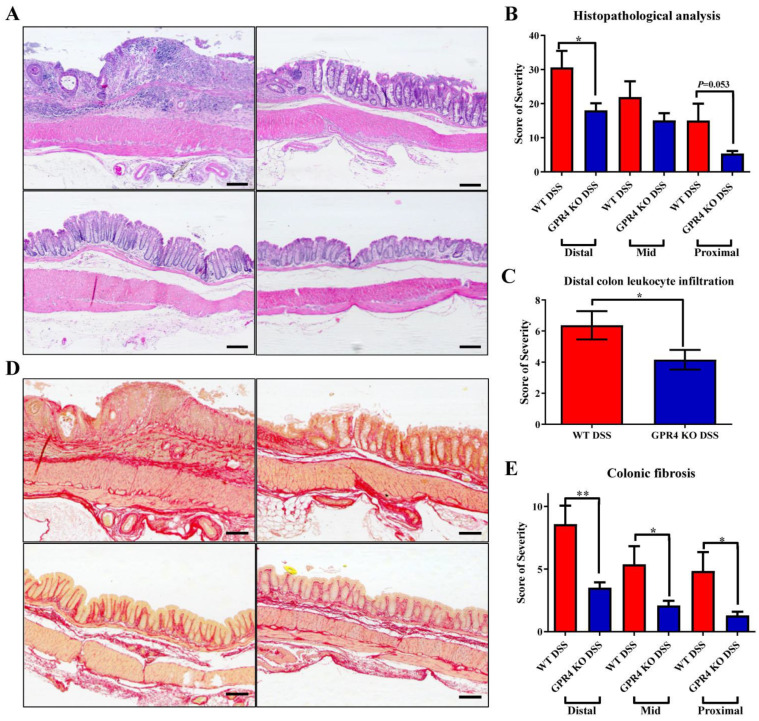
Histopathological analysis of proximal, middle, and distal colon in chronic colitis mice. Characteristic histopathological features of colitis were assessed to further characterize the degree of intestinal inflammation. WT-DSS mice presented elevated disease indicators compared with GPR4 KO-DSS mice. Representative H&E pictures of distant colon were taken for (**A**) WT-DSS, GPR4 KO-DSS, WT control, and GPR4 KO control mice. (**B**) Total histopathologic scores for WT-DSS and GPR4 KO-DSS proximal, middle, and distal colons. For fibrosis assessment, representative pictures of Picrosirius red-stained tissue sections of distant colon were taken for (**D**) WT-DSS, GPR4 KO-DSS, WT-control, and GPR4 KO control mice. Graphical representations of (**B**) total histopathological parameters, (**C**) leukocyte infiltration assessment in the distal colons, and (**E**) colonic fibrosis are presented. WT control (n = 18), WT-DSS (n = 17), GPR4 KO control (n = 14), and GPR4 KO-DSS (n = 18) mouse tissues were used for histopathological analysis. Scale bar is 100 µm. Data are presented as the mean ± SEM, and statistical significance was determined using an unpaired *t*-test between the WT-DSS and GPR4 KO-DSS groups (* *p* < 0.05, ** *p* < 0.01).

**Figure 3 cancers-15-04974-f003:**
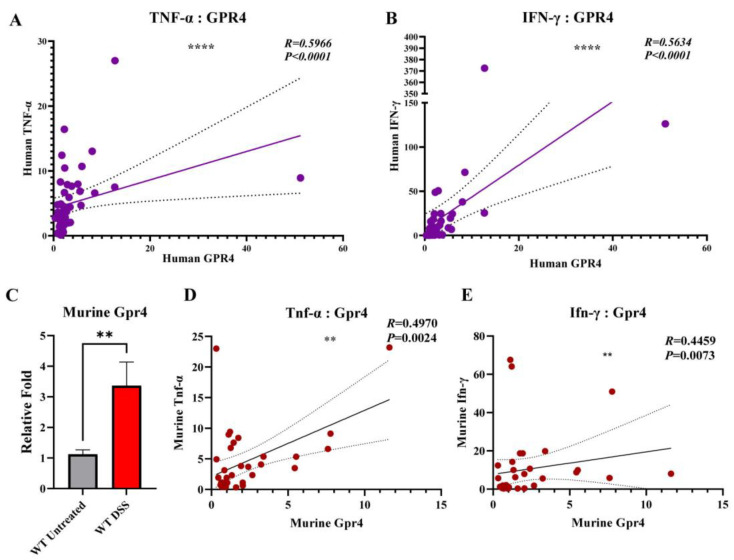
GPR4 gene expression in the inflamed colon tissues and correlation with inflammatory mediators. GPR4 gene expression is correlated with inflammatory mediator TNF-α and INF-γ gene expression in colon tissue samples of IBD patients. A positive correlation was found between (**A**) GPR4 and TNF-α and (**B**) GPR4 and IFN-γ relative mRNA expression in IBD patient samples. Statistical significance was tested with the Pearson correlation coefficient. Dashed lines indicate the 95% confidence band of the best-fit line with the linear regression analysis. (**C**) Upregulation of murine GPR4 gene expression in WT-DSS inflamed colon tissues compared with the control. Correlation between (**D**) murine GPR4 and TNF-α and (**E**) Gpr4 and Ifn-γ relative mRNA expression in WT-DSS and control mouse colon samples. Statistical significance was tested using Pearson or Spearman correlation coefficient (** *p* < 0.01, **** *p* < 0.0001).

**Figure 4 cancers-15-04974-f004:**
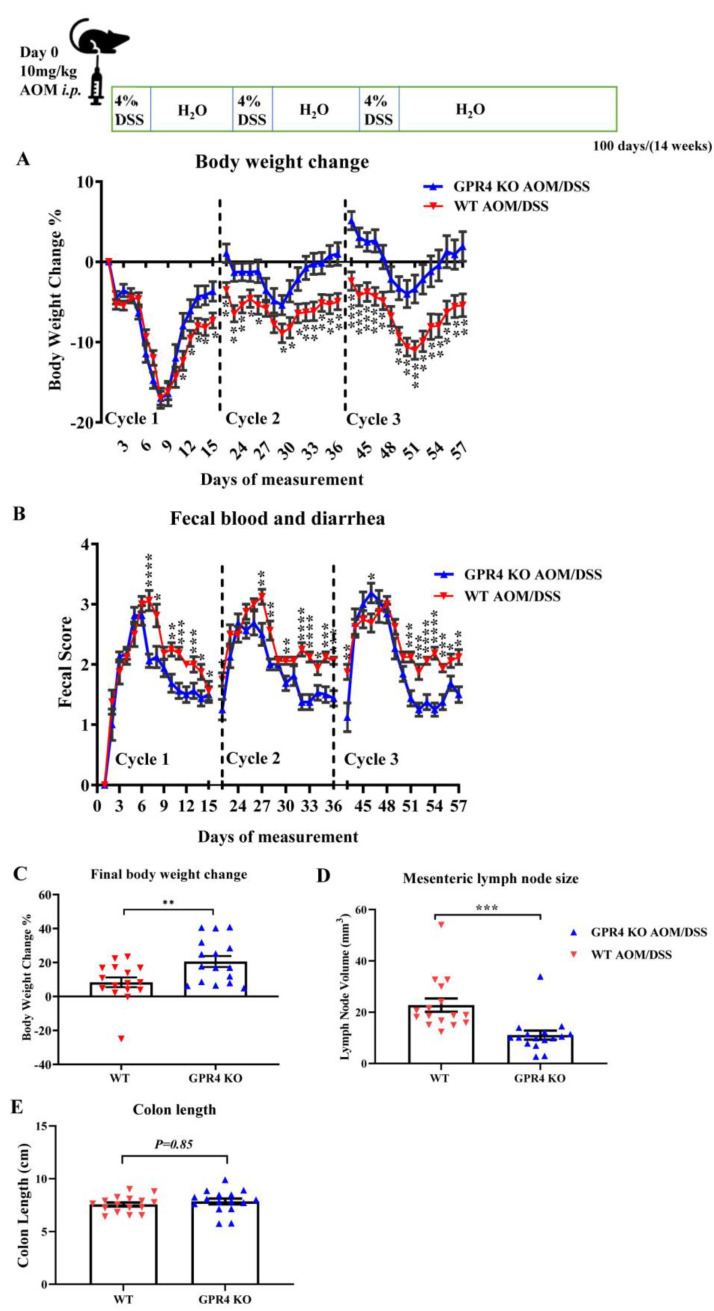
GPR4 knockout reduces intestinal inflammation in the colitis-associated colorectal cancer (CAC) mouse model. WT AOM/DSS (n = 16) and GPR4 KO AOM/DSS (n = 16) mice were used for this analysis. WT AOM/DSS mice showed more severe body weight loss and fecal blood and diarrhea scores throughout the treatment cycles when compared with GPR4 KO AOM/DSS mice. (**A**) Body weight change percentage and (**B**) fecal blood and diarrhea score. At the endpoint, disease parameters such as final body weight change, colon length, and mesenteric lymph node expansion were measured. WT AOM/DSS mice showed less body weight gain and increased mesenteric lymph node expansion when compared with GPR4 KO AOM/DSS mice. (**C**) Final body weight change percentage, (**D**) mesenteric lymph node size, and (**E**) colon length. Data are presented as the mean ± SEM. Statistical significance was determined using an unpaired *t*-test between WT AOM/DSS and GPR4 KO AOM/DSS mice (* *p* < 0.05, ** *p* < 0.01, *** *p* < 0.001, and **** *p* < 0.0001).

**Figure 5 cancers-15-04974-f005:**
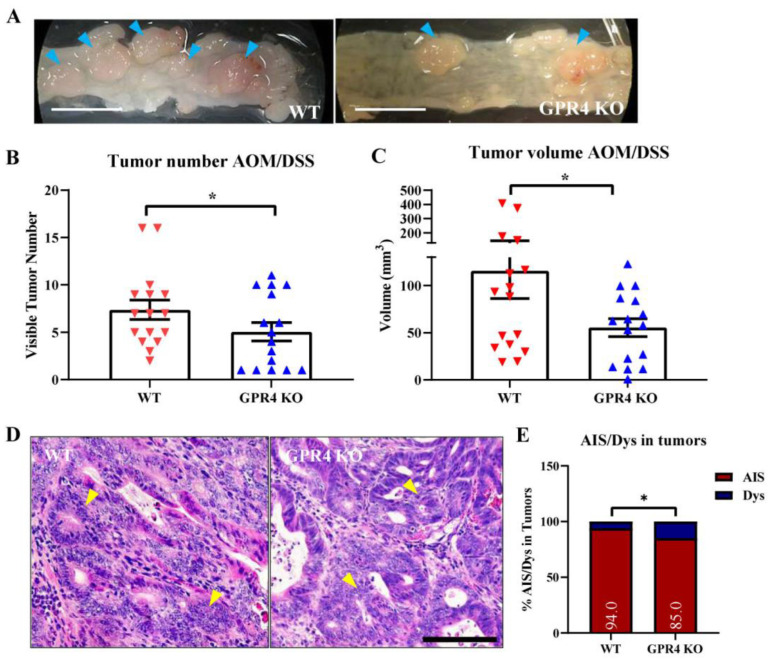
Effects of GPR4 on tumorigenesis in the colitis-associated colorectal cancer (CAC) mouse model. Tumor burden was evaluated using clinical observation of tumor number and tumor volume at the endpoint for WT AOM/DSS (n = 16) mouse colons compared with GPR4 KO AOM/DSS mouse colons (n = 16). Tumor burden was found to be higher in WT AOM/DSS mouse colons compared with GPR4 KO AOM/DSS mouse colons. (**A**) Representative pictures of distal colons bearing tumors (indicated by blue arrowheads), (**B**) tumor number, (**C**) tumor volume, (**D**) histopathological representation of adenocarcinoma in AOM/DSS mouse colons (yellow arrowheads), and (**E**) distribution of adenocarcinoma in situ (AIS) and low-grade dysplasia (Dys) in WT AOM/DSS and GPR4 KO AOM/DSS mouse tumors. Data are presented as the mean ± SEM, and statistical significance was determined using an unpaired *t*-test and a chi-square test between WT AOM/DSS and GPR4 KO AOM/DSS mice (* *p* < 0.05). Scale bar in (**A**) is 1cm, and scale bar in (**D**) is 100 µm.

**Figure 6 cancers-15-04974-f006:**
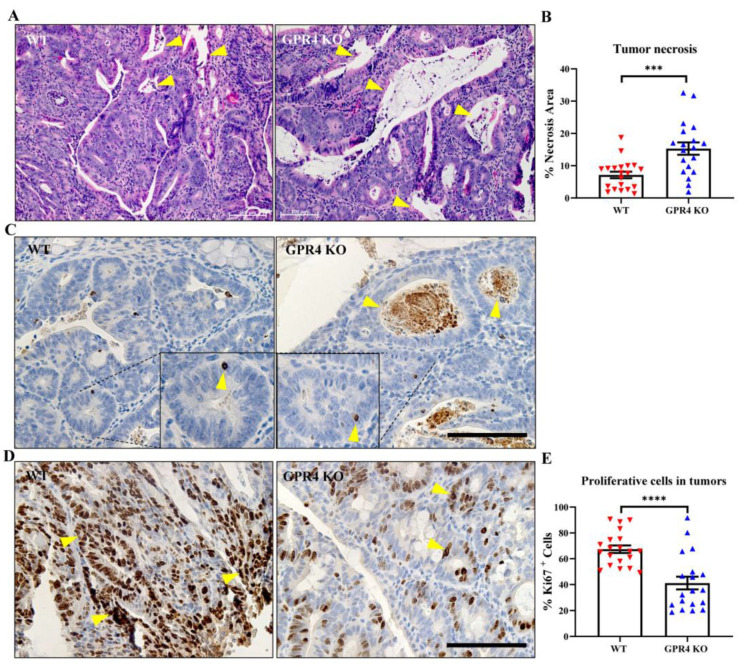
Tumor necrosis, cell death, and proliferation in the tumor tissues of the colitis-associated colorectal cancer (CAC) mouse model. H&E stain was used to assess necrotic areas in WT AOM/DSS (n = 20 tumors) and GPR4 KO AOM/DSS (n = 19 tumors) mouse tumors. GPR4 KO AOM/DSS showed higher percentage of tumor necrosis when compared with WT AOM/DSS. (**A**) Representative images of H&E-stained tumor tissues showing necrotic areas (yellow arrowheads) and (**B**) quantification of average percent of tumor necrosis area in WT AOM/DSS and GPR4 AOM/DSS mouse tumors. Cell death was assessed using cleaved caspase-3 immunohistochemistry in the tumor tissues of WT AOM/DSS and GPR4 KO AOM/DSS mice. (**C**) Representative images of cleaved caspase-3+ cells (yellow arrowheads) in WT AOM/DSS and GPR4 AOM/DSS mouse tumor tissues. Ki67 + IHC was used to assess cell proliferation in WT AOM/DSS (n = 20 tumors) and GPR4 KO AOM/DSS (n = 19 tumors) mouse tumors. WT AOM/DSS tumors showed an increased percentage of proliferative cells when compared with GPR4 KO AOM/DSS tumors. (**D**) Representative images of Ki67+ (yellow arrowheads)-stained tumor tissues for WT AOM/DSS and GPR4 KO AOM/DSS mice and (**E**) quantification of average percent of Ki67+ proliferative cells in WT AOM/DSS and GPR4 AOM/DSS mouse colon tumors. Images were quantified using the Fiji (v.1.53q) software. Data are presented as the mean ± SEM, and statistical significance was determined using the unpaired *t*-test (*** *p* < 0.001, **** *p* < 0.0001). Scale bar is 100 µm.

**Figure 7 cancers-15-04974-f007:**
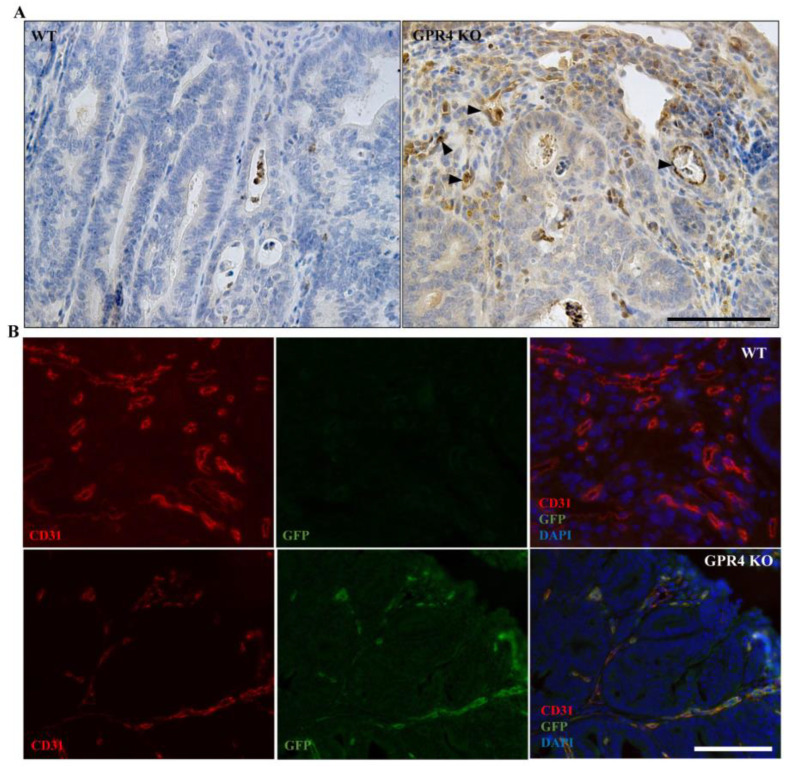
GFP signal and double labeling with CD31 in the colon tumor tissues of the colitis-associated colorectal cancer (CAC) mouse model. GFP knock-in under the control of the GPR4 promoter serves as a surrogate marker for endogenous GPR4 expression in GPR4 KO mice. GFP expression could be detected via immunohistochemistry in GPR4 KO mice but not in WT mice. (**A**) GFP expression (indicated by black arrowheads) in GPR4 KO AOM/DSS tumors versus no expression in WT AOM/DSS tumors and (**B**) double labeling of GFP (green) and CD31 (red). Blood vessels show double-positive signals for GFP and CD31 in the GPR4 KO AOM/DSS tumors versus only a CD31-positive signal in the WT AOM/DSS tumors. Scale bar is 100 µm.

**Figure 8 cancers-15-04974-f008:**
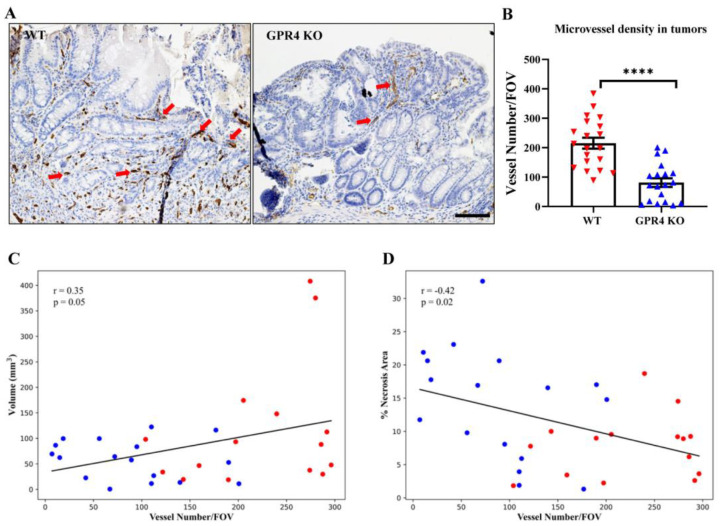
Microvessel density in the tumor tissues of the colitis-associated colorectal cancer (CAC) mouse model. Using CD31 immunohistochemistry, blood vessel numbers were assessed in WT AOM/DSS mice (n = 20 tumors) and GPR4 KO AOM/DSS mice (n = 19 tumors). WT AOM/DSS tumors showed an increased number of blood vessels compared with GPR4 KO AOM/DSS tumors. (**A**) Representative pictures of WT and GPR4 KO AOM/DSS tumor CD31+ blood vessels (red arrows); (**B**) quantification of average microvessel density per field of view (FOV) in the tumors; (**C**) positive correlation between tumor blood vessel density and tumor volume in WT AOM/DSS and GPR4 KO AOM/DSS mouse colons; and (**D**) negative correlation between tumor blood vessel density and tumor necrosis area in WT AOM/DSS and GPR4 KO AOM/DSS mouse colons. Red and blue dots denote WT AOM/DSS and GPR4 KO AOM/DSS tumors, respectively. ImageJ was used for quantification. Data are presented as the mean ± SEM, and statistical significance was determined using the unpaired *t*-test (**** *p* < 0.0001). Scale bar is 100 µm.

## Data Availability

All data are included within the manuscript and available from the corresponding author upon request.

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
