# Peer review of "GPR4 Knockout Attenuates Intestinal Inflammation and Forestalls the Development of Colitis-Associated Colorectal Cancer in Murine Models"

_cancers, 2023, doi:10.3390/cancers15204974_

Round 1

Reviewer 1 Report (Previous Reviewer 1)

Thank you for your correction.

Author Response

Reviewer #1

Thank you for your correction.

Response: Thank you for your constructive comments that helped improve this manuscript.

Reviewer 2 Report (Previous Reviewer 2)

The authors have adequately addressed my suggestions. This version of the manuscript is significantly improved.

Minor comment: Gene name should be in italics.

Author Response

Reviewer #2

The authors have adequately addressed my suggestions. This version of the manuscript is significantly improved.

Minor comment: Gene name should be in italics.

Response: Thank you for the suggestion. We have changed gene names in italics.   

Reviewer 3 Report (Previous Reviewer 3)

 GPR4 knockout attenuates intestinal inflammation and forestalls the development of colitis-associated colorectal cancer in murine models, is a resubmission of paper withdrawn in June of July of 2023.  In the interim, the authors added DSS only in the study as recommended by this reviewer in the first round of review.

The current version also includes correlations of vessel density and tumor volume and necrosis that I wanted to see.

As such, most of my previous comments have been addressed.  The only outstanding issue, and this might affect other reviews of this version, is the sloppy presentation that incorporates revisions.  Several figures seem to sit over other figures and fig. 8 is a real mess, being split into 2 parts with intervening text.   

Minor-line 77 vs. 89-backcross generations-why 2 different values?

Hard to tell with so many edits in place.

Author Response

Reviewer #3

GPR4 knockout attenuates intestinal inflammation and forestalls the development of colitis-associated colorectal cancer in murine models, is a resubmission of paper withdrawn in June of July of 2023.  In the interim, the authors added DSS only in the study as recommended by this reviewer in the first round of review.

The current version also includes correlations of vessel density and tumor volume and necrosis that I wanted to see.

As such, most of my previous comments have been addressed.  The only outstanding issue, and this might affect other reviews of this version, is the sloppy presentation that incorporates revisions.  Several figures seem to sit over other figures and fig. 8 is a real mess, being split into 2 parts with intervening text.   

Response: This issue was due to the track of changes in Word during the manuscript revision process. We apologize for the formatting problem. We have included a clean version without the track of changes.

Minor-line 77 vs. 89-backcross generations-why 2 different values?

Response: The knockout mouse colonies in our lab are backcrossed periodically to mitigate the impact of potential genetic drift. The mice used in the DSS experiment were backcrossed for 11 generations. And the mice used in the AOM/DSS experiment were backcrossed for 13 generations.

Round 2

Reviewer 3 Report (Previous Reviewer 3)

The authors have addressed my concerns adequately.

This manuscript is a resubmission of an earlier submission. The following is a list of the peer review reports and author responses from that submission.

Round 1

Reviewer 1 Report

This study seems to be attractive, but some data need revision.

Fig2A should show the same regions corresponding to Fig2D. Authors should reselect suitable pictures for "WT DSS" and "GPR4 KO DSS" in Fig2A.

The result of TNF-alpha is important but unclear in this manuscript. Authors should show pictures with same morphology and with same intensity-level of background, in Fig3C. Please reselect suitable pictures for Fig3C. In addition, authors should perform western blotting analysis of TNF-alpha, using tissue samples or blood samples.

Moreover, section 3.5. and Figure 6 may be wrong. I will reconsider after revision.

Reviewer 2 Report

In this manuscript, the authors investigated the role of GPR4, a proton-sensing GPCR, in colitis and colitis-associated cancer using murine models. The role of GPR4 in colonic inflammation and colon carcinogenesis have been studied earlier using DSS-induced acute mice model or orthotopic tumor mice model, respectively. However, in this manuscript authors studied the role of GPR4 in chronic colonic inflammation and CAC by using chronic DSS model and AOM/DSS model of CAC, respectively. They show that deficiency of GPR4 alleviates intestinal inflammation, tumor angiogenesis and CAC in mice, yet the mechanistic insight revealed in this study remains limited. Although, authors concluded decreased inflammatory mediators and microvessels density in tumors as probable mechanisms of reduced colitis and CAC in GPRKO mice, the direct evidence is lacking. I offer few comments and suggestions for improvement as below:

-       Fig. 1, effect of GPR4 on chronic colitis is marginal. Substantiating these data with fecal lipocalin level in WT DSS and GPR4 KO DSS will strengthen the conclusion.

-       Fig 2, histopathological scoring shows reduced infiltration of leukocyte in distal colon, that should be further validated by immunostaining for CD45. Also characterizing the specific immune cell type, if any, for the observed phenotype is important.

-       Fig 2A, a representative histology picture of which part of colon is presented, should be mentioned. Also, it would be good to add representative histology pictures of other colonic parts as supplementary figures in addition to histopathological analysis given in Fig. 2B.

-       Criteria of differentiating dysplasia with adenocarcinoma in histological sections should be mentioned.

-       It is not clear which cells in the colon are primarily undergoing apoptosis and contributing to decreased CAC. Cleaved casp-3 staining is primarily seen in tumor necrotic area (Fig. 8C) and no. of microvessels are significantly less in GPR4KO (Fig. 7A). Just wondering if CD31+ endothelial cells are undergoing apoptosis resulting reduced microvessels, or intestinal epithelial cells (IECs) are undergoing apoptosis in absence of GPR4. It would be good to do immunofluorescence co-staining for CD31 with cleaved casp-3, and also for Epcam (IECs marker) with cleaved casp-3.

-        Why different DSS concentrations were used for chronic DSS mice model and AOM/DSS CAC mice model? Also, what was the basis of giving just 2 days of recovery time between the DSS cycles in chronic DSS model in oppose to 14 days of recovery as widely used?

-       There are a few typos e.g., as in line 193, diahrea.

Manuscript is written in good English except few typos.

Reviewer 3 Report

The paper, GPR4 knockout attenuates intestinal inflammation and forestalls the development of colitis-associated colorectal cancer in murine models, examines the role of the gene and associated G-protein in colon inflammation and combination carcinogen and inflammation tumorigenesis.  For the most part this is a straightforward observational study supporting a significant role of GPR4 in inflammation.  While I can appreciate the desire to link this to tumorigenesis due to the association between IBD and colon cancer, I wish that the authors had considered a different tact when it came to the use of DSS and azoxymethane.

As I am sure you are aware, DSS (Ishioka T, Kuwabara N, Oohashi Y, Wakabayashi K. Induction of colorectal tumors in rats by sulfated polysaccharides. Crit Rev Toxicol. 1987;17(3):215-44. doi: 10.3109/10408448709071209. PMID: 2438086) and azoxymethane or DMH (Ward JM, Yamamoto RS, Brown CA. Pathology of intestinal neoplasms and other lesions in rats exposed to azoxymethane. J Natl Cancer Inst. 1973 Sep;51(3):1029-39. doi: 10.1093/jnci/51.3.1029. PMID: 4355212) can be used separately to induce colon tumors.   It would have been a cleaner study to have used DSS alone to induce the tumors and then separately used AOM. You did, what you did, and I must work with that.  My reason for wanting AOM alone is to have case where inflammation is not a major factor and the angiogenesis contribution to tumor volume and necrosis would be clearer.  There is probably going to be some inflammation with AOM, but much less than with the combination. My wanting DSS alone follows a similar vein. There, I could see just how much GPR4 acts in the inflammatory role to contribute to tumorigenesis without the unnecessary baggage of a carcinogen.

Working with what was provided, I suggest a few alterations. In Fig.2 a specify which segment we are seeing in the picture: the panel set in D is a little unclear to me.  Can better images be supplied maybe at higher magnification and again specify the segment. Fig 3 panels a and b just upset the flow of the paper and as pointed out in the text represents information that could be elsewhere inferred (panel c) or already known in the literature. Figs 3 c and d just seems like a continuation of Fig 2 and could be consolidated there. What I would rather see in lieu of Fig 3 is how GPR4 (or GPR4 protein)levels compare in Wt mice with and without DSS.

From the data shown in Figures  5 and 7, is there a correlation between the microvessel density and tumor volume and any inverse correlation for microvessel density and necrosis?